# Management of *Spodoptera frugiperda* J.E. Smith Using Recycled Virus Inoculum from Larvae Treated with Baculovirus under Field Conditions

**DOI:** 10.3390/insects14080686

**Published:** 2023-08-03

**Authors:** Allan Mweke, Ivan Rwomushana, Arthur Okello, Duncan Chacha, Jingfei Guo, Belinda Luke

**Affiliations:** 1Department of Animal Health and Production, School of Pure and Applied Sciences, Mount Kenya University, Thika P.O. Box 342-01000, Kenya; 2Centre for Agriculture and Biosciences International (CABI) Africa, Canary Bird, 673 Limuru Road, Muthaiga, Nairobi P.O. Box 633-00621, Kenya; 3Institute of Plant Protection, Chinese Academy of Agricultural Sciences (IPP-CAAS), Beijing 100193, China; guojingfei1989@126.com; 4Centre for Agriculture and Biosciences International (CABI), Bakeham Lane, Egham, Surrey TW20 9TY, UK; b.luke@cabi.org

**Keywords:** fall armyworm, damage, Littovir, insecticides, maize, yield, virus

## Abstract

**Simple Summary:**

Maize is the staple food in Sub-Saharan Africa and a source of livelihood for millions of smallholder farmers. Constraints in production, which include pest damage, lead to loss of production and hunger. Since its arrival in Kenya in 2016, fall armyworm (FAW) has caused huge destruction. Chemical control is the preferred choice by farmers despite its negative effects. Baculoviruses offer a sustainable alternative to pesticides. However, their cost and special storage requirements make them unattractive to smallholder farmers, especially where repeat applications are required. The potential to use recycled virus inoculum from FAW larvae treated with a commercial baculovirus product has not been documented. This study evaluated the efficacy of recycled virus inoculum from larvae treated with Littovir, a commercial product, under laboratory and field conditions. Under laboratory conditions, the recycled virus inoculum caused varying mortality in different FAW instars, with the highest mortality recorded in the 1^st^–3^rd^ instars. Under field conditions, the recycled virus inoculum produced maize yield comparable to that of commercial insecticides but similar to that of the control. This study has highlighted the potential of recycled virus inoculum from larvae treated with a commercial product. This approach offers affordable means of controlling FAW since farmers only need to purchase the commercial product once and can use recycled virus inoculum from treated larvae for repeat applications.

**Abstract:**

Fall armyworm (FAW) is a major pest of maize and causes huge losses. Chemical pesticides are the commonly used control strategy among farmers. The efficacy of baculoviruses against FAW has been proven; however, farmers may not be able to afford the products. The use of farmer-produced baculovirus mixtures could provide an opportunity for a nature-based solution for FAW at a low cost. This study evaluated the potential of recycled virus inoculum from FAW larvae treated with a commercial baculovirus (Littovir) for the management of FAW under laboratory and field conditions. In the laboratory, the virus from 25, 50, 75 and 100 FAW larvae caused variable mortality among FAW instars. The highest mortality (45%) among 1^st^–3^rd^ instars was caused by Littovir followed by recycled virus inoculum from 100 FAW larvae (36%). Under field conditions, even though recycled virus inoculum did not offer adequate protection against FAW damage, the maize yield was comparable to that of commercial insecticide-treated plots and similar to that of control plots. This study has shown the potential use of recycled virus inoculum from infected larvae for the management of FAW. This would offer the farmers a sustainable and affordable option for the management of FAW as it would require the farmers to purchase the commercial baculovirus once and collect larvae from treated plots for repeat applications.

## 1. Introduction

Maize (*Zea mays* L.) is the main staple food crop in sub-Saharan Africa (SSA) and is mainly grown by smallholder farmers [1,2]. The smallholder, resource-poor farmers produce mainly for subsistence, and they require an affordable and sustainable production system to feed themselves. Maize pests pose the greatest challenge to productivity among smallholder farmers, and fall armyworm (FAW), *Spodoptera frugiperda* (J.E. Smith) (Lepidoptera: Noctuidae), is one of the most destructive pests [3,4]. The pest is polyphagous and, since its arrival in Africa, has left destroyed millions of hectares of maize plantations with economic losses estimated at USD 9.6 bn [5,6,7]. A number of management and adaptation strategies, including hand picking, crop rotation, early planting, application of soil or wood ash in the whorls and planting early-maturing crops, have been applied to reduce damage and crop loss [8,9]. However, chemical control remains popular among farmers and produces better grain yields compared to other control strategies [10,11]. In Kenya, a number of insecticides have been registered for use against FAW since its invasion. These include Diazinon, Alpha Cypermethrin, Chlorpyrfos, Diflubenzuron Triclorfon (Dipterex), Chlorantraniliprole, Spinetoram, Emamectin benzoate, Indoxacarb and Lambda Cyhalothrin (https://www.pcpb.go.ke/crops/ (accessed on 10 June 2022). Despite their popularity, pesticides have negative and undesirable effects, including human health effects, food safety concerns emanating from pesticide residues, and development of resistance by the pest, and they also affect beneficial non-target organisms [12,13]. The potential of the use of baculoviruses, particularly the *S. frugiperda* multiple nucleopolyhedrovirus (SfMNPV), has been demonstrated, and some commercial products are available in various countries, including Kenya [14,15,16,17]. Integrated pest management and use of microbial-based biopesticides like baculoviruses is a good alternative to synthetic insecticides. The baculoviruses have a number of advantages that include specificity and low/no health risks, are simple to apply, can be formulated in a number of ways and do not harm beneficial non-target organisms [18,19]. The efficacy of baculoviruses against FAW and other lepidopterans has been reported in several studies [20,21]. 

The baculovirus infection starts when the FAW ingests occlusion bodies (OBs) on the treated leaf surface. The midgut of the larvae is alkaline and dissolves the OBs, releasing the virions that bind to, and infect, the epithelial cells of the midgut after crossing the peritrophic membrane (PM) [22]. The infected midgut cells produce a second virus phenotype, named the budded virus (BV), which causes systemic infection [22]. Research has shown that the efficiency of a baculovirus is influenced by the type of baculovirus formulation [22].

A limitation of baculoviruses is that they do not cause instantaneous death of the pest. Mortality is observed days after applications; however, this is not a major challenge in maize because the crop can withstand moderate defoliation without compromising yield [19]. Studies on the effect of FAW defoliation on grain yield have shown that the damage caused by the pest has a minimal but detectable impact on maize yield [22]. Additionally, some baculovirus products are expensive, and smallholder farmers may not be able to afford repeated applications that enhance efficacy. The solution to this problem may be an alternative control option that is affordable for the resource-poor smallholder farmers.

Therefore, this study evaluated the efficacy of recycled virus inoculum from infected FAW larvae sprayed with a commercial baculovirus product, to test an approach that would reduce the cost of FAW control by smallholder farmers [18]. In this approach, farmers would only need to buy the commercial product once and use recycled virus inoculum from treated larvae for subsequent applications. The virus product used in this study (Littovir) has been tested in a number of African countries against FAW and is registered in Cameroon, Morocco and Tunisia [20]. Littovir contains a *Spodoptera littoralis nucleopolyhedrovirus* (SpliNPV) as the active ingredient.

## 2. Materials and Methods

### 2.1. Study Site and Experimental Plots

The experiment was carried out at Kenya Agricultural and Livestock Research Organization (KALRO) Muguga, Kiambu County, between August 2020 and October 2021. The station lies at an altitude of 1675 m above sea level, longitude 36.6579649 E and latitude −1.2551409 S. The laboratory experiments were carried out between August 2020 and February 2021, while the field experiments were carried out between March and October 2021. Under field conditions, treatment applications were applied between May and June 2021. During the experimental period, the mean daily temperature was 21 °C, and the mean annual precipitation was 136.6 mm.

### 2.2. Maize Seedlings

Eight maize seeds, variety H514, per pot were planted in pots measuring 17 cm (diameter) and 17 cm (depth) and placed in the open field to allow germination, after which they were thinned to 4 seedlings per pot. After 7 days post-germination, the seedlings were transferred to a greenhouse to avoid infestation by FAW and other pests. They were regularly watered and used to feed FAW larvae.

### 2.3. Fall Armyworm Colony

The initial FAW colony was established by collecting FAW larvae from infested maize plants, in the open field at KALRO Muguga, and then transferred to the laboratory for rearing. FAW larvae were transferred, using a soft camel brush to pick 1^st^ to 3^rd^ instars and soft forceps to pick larger instars (4^th^–6^th^), into boxes (22 cm (length) and 15 cm (width)) lined with a paper towel to absorb moisture from the maize leaves. The boxes had tops with fine nets to allow for air circulation. The net apertures were small enough to prevent larvae from escaping. The FAW larvae were fed with cultivated maize leaves of variety H514. The maize leaves were harvested and cleaned with water; then, they were air-dried for 10 min to remove excess moisture before being fed to the larvae. The larvae were fed every 2 days. The mean temperature in the FAW rearing room was 28 °C, and RH was 80%. A thermostatic heater was used to maintain favorable temperatures for FAW growth and development since the night temperatures could drop to as low as 8 °C.

### 2.4. Laboratory Evaluation of Efficacy of Recycled Virus Inoculum from FAW Larvae Sprayed with Littovir

#### 2.4.1. Inoculum Preparation

(a)Initial inoculum treatment with baculovirus (Littovir)

The initial inoculum was prepared by suspending 5 × 10^11^ OB/L and left for 10 min. Maize seedlings were prepared by cutting from the base and cleaning with tap water. Excess water was removed using paper towels. They were then immersed in a basin containing 5 × 10^11^ OB/L Littovir suspension for 5 min. Subsequently, they were air-dried for 10 min to remove excess moisture and transferred into the aerated boxes. Twenty (20) FAW larvae (1^st^–6^th^ instars) were then transferred into the containers and kept at room temperature. The different larval instars were evaluated separately. Maize seedlings were replaced regularly to avoid cannibalism among the larvae. Mortality was recorded daily for 7 days, and dead larvae were collected and stored in a fridge at 4 °C for later use. This treatment had 4 replications to obtain enough infected larvae to use in the bioassays. In the laboratory experiment, FAW instars were separated into two groups, i.e., 1^st^–3^rd^ instars and 4^th^–6^th^ instars (to avoid cannibalism of the younger instars by the older ones), before being subjected to treatment. Each group contained 30–40 larvae. 

(b)Virus from FAW larvae treated with baculovirus (Littovir)

When preparing recycled virus inoculum from FAW larvae, the larvae were first placed in the freezer at 4 °C for 20 min to immobilize them. The larvae were then transferred into a glass vial and crushed, using a pestle, into a paste. The dead larvae were crushed and then suspended in 10 mL of tap water to form a suspension.

#### 2.4.2. Treatment

There were 6 treatments, as follows: (1) Littovir, (2) untreated control, (3) recycled virus inoculum from 25 larvae, (4) recycled virus inoculum from 50 larvae, (5) recycled virus inoculum from 75 larvae and (6) recycled virus inoculum from 100 larvae. Twenty (20) FAW larvae (1^st^–6^th^ instars) were used in each treatment, and each treatment was replicated 3 times.

### 2.5. Field Evaluation of Efficacy of Recycled Virus Inoculum from FAW Larvae Sprayed with a Commercial Baculovirus Product

#### 2.5.1. Initial Inoculum for Field Evaluation and Seed Bed Preparation

Land was plowed and harrowed 1 month before the start of the rainy season. Leveling was performed to ensure the land was even. A plot measuring 30 × 50 m was planted earlier than the experimental plots. Maize was planted at 75 cm inter-row by 25 cm interplant spacing. All agronomic practices for maize production were carried out according to recommendations. The plants, in the field, were naturally infested by FAW. After infestation, the plants were sprayed with Littovir at the recommended rate of 5 × 10^11^ OB/L. This was the source of initial inoculum for recycling of the virus from FAW larvae for field treatments.

#### 2.5.2. Engeo 247SC (Sygenta) and Escort 1.9EC (Green Life Crop Protection)

The insecticides used as the positive control in this study are commonly used for the control and management of FAW in Kenya. Engeo (141 g/L Thiamethoxam and 106 g/L Lambda-cyhalothrin) was applied at the recommended rate of 8 mL/20 L of water or 150 mL/Ha in 500 L of water, while Escort (Emamectin benzoate 19 g/L) was applied at the recommended rate of 25 mL/20 L of water or 500 mL/ha in 400 L of water. The pesticides were thoroughly mixed with tap water prior to application and applied using separate knapsack sprayers to avoid contamination.

#### 2.5.3. Soil and Weather Conditions in Muguga

The Muguga area has an average temperature of 16 °C with daily temperatures rarely exceeding 28 °C or falling below 8 °C (Muguga Meteorological Station). Muguga is near the equator; therefore, there are minimal day-length variations. The Muguga area has gently sloping hills and well-drained clay-loam soils. The fertile soils in the area are originally from lava and are generally very deep [23].

#### 2.5.4. Crop

Maize variety H614, acquired from local agrodealers, was planted in plots measuring 10 × 10 m. Two seeds per hole were planted and allowed to germinate. The intra-plant and inter-row spacing was 30 cm by 75 cm, respectively. After germination, thinning was performed to leave 1 seedling per hole. The total plant population was 333 plants per plot. All the agronomic practices were carried out safe for the application of pesticides. The experiment was carried out between April and October 2021. The treatment application was performed between May and June 2021.

#### 2.5.5. Treatments, Layout and Design

Following the laboratory experimental results where recycled virus inoculum from 25, 50 and 75 larvae caused low mortalities, they were dropped from the field experiment, and recycled virus inocula from 100, 150, 200 and 250 FAW larvae were used instead. There were 8 treatments, as follows: (1) Littovir, (2) untreated control, (3) Engeo insecticide, (4) Escort insecticide, (5) recycled virus inoculum from 100 FAW larvae, (6) recycled virus inoculum from 150 FAW larvae, (7) recycled virus inoculum from 200 FAW larvae and (8) recycled virus inoculum from 250 FAW larvae. The experimental design was a completely randomized block design. Each treatment was replicated 5 times; hence, there were 40 experimental plots. Treatment applications were applied weekly for 5 weeks.

### 2.6. Evaluation of Treatments

#### 2.6.1. FAW Infestation Assessment

Immediately after maize germination, FAW pheromone traps (FAW lure from Farmtrack Consulting Ltd. (Nairobi, Kenya)) were installed at the rate of 4 traps per hectare to monitor FAW infestation. Twenty (20) maize plants per treatment were randomly selected from each experimental plot. The plants were thoroughly examined on the leaves and whorls for the presence of FAW larvae and eggs. The larvae found on the plants were counted and recorded. Sampling was carried out weekly and began 4 weeks after germination. 

#### 2.6.2. FAW Leaf Damage Assessment

Leaf damage assessment was carried out once before treatment application and thereafter once every week by assessing twenty (20) random maize plants from each experimental plot. The plants were thoroughly examined on the leaves and whorls for FAW damage. The damage was scored using the following damage scale: 1 = no damage to any ears; 2 = tip (<3 cm) damage to 1–3 ears; 3 = tip damage to 4–7 ears; 4 = tip damage to 7 or more ears and damage to 1–3 kernels below ear tips on 1 to 3 ears; 5 = tip damage to 7 or more ears and damage to 1–3 kernels of 4 to 6 ears; 6 = ear tip damage to 7–10 ears and damage to 1–4 kernels below tips of 7 to 10 ears; 7 = ear tip damage to 7–10 ears and damage to 4–6 kernels destroyed on 7–8 ears; 8 = ear tip damage to all ears and 4–6 kernels destroyed on 7–8 ears; 9 = ear tip damage to all ears and 5 or more kernels destroyed below tips of 9–10 ears [24]. Sampling was carried out weekly, 4 weeks after germination.

#### 2.6.3. Maize Grain Yield Assessment

Maize grain yield was assessed at the end of the experiment. Maize was harvested from all the experimental plots and dried to remove excess moisture. The maize cobs were then threshed using a thresher, cleaned and weighed. The yield data were then recorded. The grain yield was computed in kilograms per hectare and extrapolated to tons ha^−1^.

### 2.7. Statistical Analysis

The FAW infestation, damage assessment scores and maize grain yield were first log-transformed before the data were subjected to analysis of variance (ANOVA) and means were separated using Tukey HSD. Grain yield was expressed in kilograms per hectare and was extrapolated to tons ha^−1^. Data were analyzed using R software version 4.1.2.

## 3. Results

### 3.1. Laboratory Experiment

#### 3.1.1. Effect of Baculovirus (Littovir) on Different FAW Developmental Stages

Mortality of the 1^st^–3^rd^ instars varied significantly among the treatments (F = 12.3, d.f = 4, 18; *p* = 0.02). Littovir caused the highest mortality at 44.79%, while the recycled virus inoculum from 25 FAW larvae caused the lowest mortality at 8.3% (Table 1). However, the mortality induced by all the treatments was less than 50%, and therefore, LT_50_ was not calculated. The mortality also varied significantly across the days (7 days of data collection), F = 11.7, d.f = 4, 18; *p* < 0.001.

The mortality induced in the 4^th^–6^th^ instars was also significantly different across the different treatments (F = 9.3, d.f = 4, 18; *p* < 0.001). A similar trend where Littovir caused the highest mortality while virus inoculum caused the lowest mortality was observed (Table 2). However, the treatments induced lower mortalities in the 4^th^–6^th^ FAW instars compared to the 1^st^–3^rd^ instars. 

#### 3.1.2. Field Experiments

##### Effects of Treatments on FAW Damage to Maize Crop

The FAW damage to maize showed significant variation across the different treatments (F = 11.01, d.f = 7, 35; *p* < 0.001) (Table 3). The damage increased with time and was less in the early weeks of infestation and continued even after treatment applications. Fall armyworm larvae damage varied significantly across the weeks (F = 5.28, d.f = 4, 17; *p* < 0.001). Among the treatments, the synthetic pesticides Escort and Engeo were more effective in protecting maize against damage by FAW compared to the baculovirus (Littovir) and the recycled virus inocula from 100, 150, 200 and 250 FAW larvae (Table 3). There was no significant difference between damage to maize crops treated with Littovir, recycled virus inocula from 100, 150, 200 and 250 FAW larvae, and untreated control (Table 3). 

##### Maize Grain Yield

The only significant difference in maize grain yield obtained at the end of the experiment was between Escort and the non-treated control (F = 2.7, d.f = 7, 35; *p* = 0.02) (Table 4). Escort produced the highest yield at 4.38 tons ha^−1^. Escort produced 1.8 times more yield than the untreated control. Among the baculovirus treatments, Littovir produced about 3.1 tons ha^−1^, even though there was no significant difference in the yield among the baculovirus and the recycled virus inoculum treatments. Additionally, none of the virus treatments produced a better yield than the untreated control.

## 4. Discussion

Management of FAW relies mostly on the use of synthetic chemical pesticides which are associated with health and environmental risks, and high cost, especially for resource-poor smallholder farmers. It is therefore important to develop affordable and sustainable strategies for the management of FAW. This study evaluated the potential of using recycled virus inoculum from larvae treated with a commercial baculovirus (Littovir) under laboratory and field conditions. In the laboratory, both the Littovir and recycled virus inoculum, from FAW larvae, induced mortality, with the highest mortality induced by Littovir followed by recycled virus inoculum from 100 FAW larvae. The induced mortality was higher in the 1^st^–3^rd^ instars compared to the 4^th^–6^th^ instars even though the mortality was below 50% in all the treatments. This variation may have resulted from the fact that the young instars have not developed defense mechanisms and hence are more susceptible to virus infection. Under field conditions, inducing infections of FAW resulting in dead caterpillars is an important source of inocula for the occurrence and maintenance of epizootics [15,16]. Epizootics are desirable in biological control because the dead cadavers can spread the virus to healthy non-infected populations [20].

In the field, FAW damage and yield were evaluated for eight treatments. Among the treatments, Escort (insecticide) produced the best protection against FAW damage. Littovir performed the same as Engeo (insecticide), although there was no significant difference between Littovir, the recycled virus inoculum and the untreated control. Increasing the number of treated larvae from which the virus inoculum was prepared, and used as a treatment, did not offer any additional advantages. In fact, there was no significant difference in terms of crop protection from damage to maize crops between the 100 and 250 larvae treatments. This is advantageous because farmers will spend less time collecting 100 larvae compared to 250 larvae. The baculovirus used in this experiment (Littovir (SpliNPV)) is adapted to infect *S. littoralis*, not *S. frugiperda*, and this could explain the low levels of mortality even under optimum laboratory conditions. Fall armyworm damage to maize crops has been shown to be influenced by cropping systems and agricultural practices and varies between monocrops and intercrops, with monocrops having more damage [11,25]. The maize in this study was planted as a monocrop, and there were neighboring monocrop maize fields near the trial sites. This may have increased the FAW population pressure resulting in damage. Fall armyworm pheromone traps were installed immediately after germination to monitor FAW infestation. Treatment commenced four weeks after germination, and this was to allow for the crop to attain uniform height and infestation levels. This suggests that considerable damage had happened by the time treatment application commenced, hence making it difficult to detect significant differences in damage after the treatments were applied. This implies that proper timing of application is critical in the effective management of FAW. However, the use of recycled virus inoculum recovered from infected larvae has limitations because the ability to induce mortality by the recycled inoculum may be reduced because the farmers may not differentiate NPV-infected larvae from those infected by other entomopathogens. The use of recycled inoculum from dead larvae infected with baculovirus poses a risk to humans because the larvae could harbor human pathogenic bacteria that could potentially harm farmers [26,27]. The similar performance between the insecticides and the baculovirus in protecting the crop against FAW damage could be explained by the fact that FAW larvae feed deep in the whorl of young maize plants, and hence a high volume of liquid insecticide may be required to obtain adequate penetration resulting in better protection against further damage. Additionally, the baculovirus is slow-acting, and some damage occurred between treatment application and action of the baculovirus. Additionally, maize is characterized by many functional leaves and can compensate the photosynthesis to ensure better crop yield, foliar damage notwithstanding and especially when infestation occurs at early stages of crop growth under good agronomic practices [6,25,28]. In addition to this, FAW larvae are known to shift feeding preference from leaf tissues to tassel, silk and ears, and this has an influence on the damage caused by FAW as the crop advances in age.

Maize grain yield was evaluated at the end of the season and expressed as tons ha^−1^. In this study, only the treatment with Escort produced a yield that was significantly different from that of the untreated control. The yield in all the other treatments was similar to that of the untreated control. This implies that the maize damage level influenced the grain yield. This study agrees with a previous study that showed that defoliation by FAW larvae on maize minimally influences grain yield even though the damage is detectable [24]. The Escort insecticide might have enhanced crop growth, resulting in higher maize yield [29]. Previous studies have shown that good agricultural practices like weeding and nutrition management enable maize to compensate for the FAW damage and produce optimum yield [30,31,32,33].

The lack of significant differences in the maize grain yield between Engeo (insecticide) and the baculovirus (Littovir) as well as the recycled virus inoculum from FAW larvae treated with Littovir demonstrates the potential of the baculovirus and the recycled virus inoculum as a sustainable FAW management strategy. This could offer resource-poor smallholder farmers a sustainable and affordable FAW management option as it negates the need for repeat applications of expensive insecticides. Previous studies have demonstrated that *Spodoptera frugiperda* multiple nucleopolyhedrovirus (SfMNPV) is highly effective in the management of FAW and is a preferred choice for biological control [16,25,34,35,36].

However, farmers should appreciate that the virus is slow-acting and does not have a knockdown effect like chemical pesticides. Hence, early application of the virus would be recommended. 

## 5. Conclusions

This study has established that Littovir recycled from dead larvae provided low mortality of *S. frugiperda* even when applied at very high rates (250 larvae). The idea of recycling was demonstrated with SpliNPV but might work considerably better with SfMNPV. The possibility that farmers could apply baculovirus once and collect larvae and recycle the virus for repeat or subsequent applications could save farmers money. The recycled virus inoculum from FAW larvae produced maize grain yield comparable to those of insecticide-treated plots, and this suggests that this approach would offer farmers benefits that include human safety, environmental protection and enhanced biological control of FAW since the virus is highly specific and does not affect non-target organisms and thus is compatible with integrated pest management strategy.

It is recommended that further studies be undertaken using natural isolates of SfMNPV in Kenya to find a more pathogenic virus. There is also a need to determine how long the farmers should keep the FAW larvae after collecting them from the field or the virus inoculum before applying a repeat application. Further studies should also be carried out to evaluate the effect of ecological parameters on the efficacy of recycled virus inocula from dead FAW larvae under different agro-ecological zones.

## Figures and Tables

**Table 1 insects-14-00686-t001:** Mean FAW 1^st^–3^rd^ instar mortality induced by different recycled virus treatments.

Treatment	FAW Instar	Mean Mortality + SE
Inoculum recovered from 25 larvae	1^st^–3^rd^	8.3 ± 1 d
Inoculum recovered from 50 larvae	1^st^–3^rd^	16.5 ± 2 dc
Inoculum recovered from 75 larvae	1^st^–3^rd^	22.5 ± 4 c
Inoculum recovered from 100 larvae	1^st^–3^rd^	36.2 ± 5 b
Littovir	1^st^–3^rd^	44.8 ± 5 a

Means followed by the same letter within a column are not significantly different by Tukey HSD at *p* < 0.05.

**Table 2 insects-14-00686-t002:** Mean FAW 4^th^–6^th^ instar mortality induced by different recycled virus treatments.

Treatment	FAW Instar	Mean Mortality + SE
Inoculum recovered from 25 larvae	4^th^–6^th^	5.2 ± 0.2 c
Inoculum recovered from 50 larvae	4^th^–6^th^	9.6 ± 0.8 bc
Inoculum recovered from 75 larvae	4^th^–6^th^	12.1 ± 1.4 bc
Inoculum recovered from 100 larvae	4^th^–6^th^	15.4 ± 1.6 ab
Littovir	4^th^–6^th^	21.7 ± 2.1 a

Means followed by the same letter within a column are not significantly different by Tukey HSD at *p* < 0.05.

**Table 3 insects-14-00686-t003:** Mean crop damage score as influenced by treatments.

Treatment	Mean Crop Damage Score + SE
Escort	1.72 ± 0.2 a
Engeo	2.00 ± 0.1 ab
Littovir	2.73 ± 0.4 bc
Inoculum recovered from 100 FAW larvae	2.80 ± 0.2 c
Inoculum recovered from 150 FAW larvae	2.80 ± 0.3 ab
Inoculum recovered from 200 FAW larvae	2.84 ± 0.3 c
Inoculum recovered from 250 FAW larvae	2.87 ± 0.4 c
Untreated control	3.30 ± 0.4 c

Means followed by the same letter within a column are not significantly different by Tukey HSD at *p* < 0.05.

**Table 4 insects-14-00686-t004:** Mean grain yield as influenced by treatments.

Treatment	Mean Grain Yield + SE (tons/ha)
Escort	4.4 ± 0.4 a
Engeo	4.0 ± 0.3 ab
Littovir	3.1 ± 0.5 ab
Inoculum recovered from 100 FAW larvae	3.0 ± 0.4 ab
Inoculum recovered from 150 FAW larvae	3.0 ± 0.4 ab
Inoculum recovered from 200 FAW larvae	3.0 ± 0.4 ab
Inoculum recovered from 250 FAW larvae	3.04 ± 0.4 ab
Untreated control	2.4 ± 0.4 b

Means followed by the same letter within a column are not significantly different by Tukey HSD at *p* < 0.05.

## Data Availability

Data can be provided upon request to the corresponding/lead author.

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
