# Peer review of "Management of *Spodoptera frugiperda* J.E. Smith Using Recycled Virus Inoculum from Larvae Treated with Baculovirus under Field Conditions"

_insects, 2023, doi:10.3390/insects14080686_

Round 1

Reviewer 1 Report

Quality of English is excellent

Author Response

The authors would really like to thank Reviewer 1 not only for accepting to review this manuscript, but also pointing out the application and relevance of the study. We have addressed Reviewer 1 concerns and revised the underlined sections as indicated below in point-by-point responses.

Comments to the Author

The manuscript evaluates the efficacy of virus extracts from Littovir-treated Spodoptera frugiperda larvae using laboratory bioassays and subsequent field trials with a view to exploring a more cost-effective approach for FAW management. The study is novel in that such an approach has not yet been investigated in the case of FAW. Furthermore, the results indicate that application of virus extracts from Littovir-treated insects may offer smallholder farmers a more affordable and sustainable option for maximising crop yield without repeated application of commercial biopesticide. The findings of the study are significant in the context of biopesticide application, and of considerable interest to subsistence farmers, commercial farmers as well as the scientific community. The manuscript is recommended for publication with minor adjustments.

Response: The authors would really like to thank Reviewer 1 not only for accepting to review this manuscript, but also pointing out the application and relevance of the study. We have addressed Reviewer 1 concerns and revised the underlined sections as indicated below in point-by-point responses.

Issue 1: The authors could comment on their choice of using Littovir and not a formulated SfMNPV for these experiments. It would be interesting to reproduce both laboratory experiments and field trials using one of the SfMNPV commercial formulations if available in Kenya.

Response: The choice of Littovir over a formulated SfMNPV for use in the experiment was because Littovir is a commercial product registered in other countries and has been evaluated for registration and therefore could be available to farmers once registered in Kenya by end of 2023. The authors thought it would not be prudent to use a product that would not be available to farmers.

Issue 2: It would be useful to provide estimates for the Littovir and virus extract inoculum concentration (OBs/ml) used in both laboratory and field experiments assuming that these were measured.

Response: The authors agree with the reviewer that it makes a lot of sense to provide Littovir and virus extract inoculum concentration (OBs/ml). However, this was not quantified during the study and therefore no information available. Additionally, the authors have provided the concentration of Littovir in the revised version of the manuscript.

Issue 3: Was molecular analysis (PCR for example) carried out to confirm the cause of death of larvae following Littovir treatment? Overt infections with the same or another virus is always possible when larvae are exposed to virus infection. Perhaps the authors could comment on the purity of the virus extract used in bioassays and field trials in the discussion (although it may not be important in the case of field-collected larvae for application).

Response: Molecular analysis was not done on dead larvae treated with Littovir and virus extract to confirm cause of death. The authors used unprurified virus extract both in the laboratory and field experiments.

Issue 4: Line 145: What was the untreated control?

Response: Untreated control is negative control.

Issue 5: Lines 228-229: Since larval mortality was calculated at different virus doses, surely LC was measured and not LT?

Response: The authors agree with the reviewer that LT instead of LC should have been calculated for different virus doses used in the bioassay. However, this was not done.

Issue 6: The authors could clarify the motivation for performing dose response bioassays on 25-100 larvae if the purpose of the study was to obtain virus extract for field trials? It is not clear how the laboratory bioassays link to the field trials especially as extract from 100-250 larvae was required for the field trials.

Response: The authors first performed laboratory bioassays with unpurified virus extracts from 25, 50, 75 and 100 larvae. However, the extracts from all except 100 larvae caused very low mortalities and the authors thought it was not prudent to use such low numbers in the field. This is why the authors increased the number of larvae and used 100, 150, 200 and 250 under field conditions. The idea was to establish the optimal number of larvae that would cause reasonable mortality. This would make the idea more appealing to the farmers since collecting huge number of larvae could be tedious and time-consuming.

Issue 7: In addition, what was the motivation for performing dose response assays on different instar stages? Perhaps the authors could comment on this in the discussion. Could the low mortality (less than 50%) be a result of using a SpliNPV isolate as opposed to a SfMNPV isolate for bioassay? The authors could further comment on this.

Response: The motivation for dose response studies on different FAW instars was to establish the most susceptible stages(instars) that can be targeted for effective control using baculovirus. This explanation has been added in the discussion. The authors also agree with the reviewer that low mortality could be due to using SpliNPV as opposed to SfMNPV isolate.

Issue 8: Lines 284-288: Please provide any relevant references to support these statements. The results are significant if the early instar stages (as opposed to later instars) are the target of pesticides/biopesticides.

Response: additional information and explanation has been added in the discussion.

Issue 9: Table 4: Correct the title (mean grain yield). The results show no difference between using virus extract from 100- 250 larvae. Does this mean that fewer larvae (100 as opposed to 250) are required to prepare the extract for field application as this would obviously be advantageous? Perhaps the authors could comment on this in the discussion.

Response: Title for table 4 has been corrected. The reviewer is right that the lack of significant difference between virus extract from 100 and 250 larvae implies it is advantageous to farmers since they would require to collect fewer larvae during treatment applications. The exercise of collecting larvae is time consuming hence farmers will require less time.

Issue 10: Line 59: remove bracket.

Response-done line 62

Issue 11: Line 142: Correct crashed to crushed.

Response-Done, line 145

Issue 12: Line 274: remove bracket.

Response: Removed line 280.

Issue 13: Line 282: correct form to from.

Response: Corrected line 284.

Reviewer 2 Report

Mweke et al., examine the use of recycled inoculum of Spodoptera littoralis nucleopolyhedrovirus (commercial product Littovir) as a means to control larvae of the fall armyworm S. frugiperda in maize in Kenya. This is an issue of importance especially for resource poor farmers in this region.
It is clear that the authors do not have great experience working with these viruses. Future studies should attempt to control for stage-related differences in larval susceptibility to the virus and should attempt to quantify virus occlusion bodies (OBs) by counting in a hemocytometer – this is not complicated and only requires a hemocytometer and a compound microscope with a x40 objective.
I have the following suggestions that should improve the manuscript.
I. The authors should attempt to provide more information on the stages of insects present in the experimental boxes. As insects inoculated in the first, second or third instars produce very little virus while an insect inoculated in the fifth instar will produce thousands of times more virus, so this is an important variable to control.
II. How much cannibalism was observed? Cannibalism especially when late instar larvae are present is a major source of larval mortality.
III. It is important to consider that when you amplify a heterologous virus (such as SpliNPV) you may recover SpliNPV or SfMNPV (through activation of a latent infection) or even a mixture of both viruses. You need to make this clear in your manuscript. You would require a genetic analysis to know what virus you have recovered in your recycled inocula.
IV. The authors need to consider the experience of soybean growers who recycled AgMNPV in Brazil but this practice was discontinued because the quality of the inoculum collected from soybean fields was too poor (as many non-infected insects or larvae that died from other diseases larvae were collected along with the AgMNPV-infected larvae) .

I have written numbered points on a scanned copy of the manuscript

Numbered points
1. Title should be improved. I suggest "Can recycled inoculum of Spodoptera littoralis nucleopolyhedrovirus be used to control Spodoptera frugiperda in Kenya?"
2. These are are NOT virus "extracts" (i.e. no extraction procedure was applied and no solvent or other extraction process was used). You need to change this terminology in the entire manuscript. I would suggest "recycled inoculum" or 'inoculum recovered from Sf larvae'.
3. Have you any idea of the amounts of virus involved in these concentrations? (25 - 100 virus killed larvae). Were you able to quantify OBs at all?
4. You need to consider the DISadvantages of recycling inoculum. Consider the experience of recycling AgMNPV in Brazil for example. What about risks from potentially human pathogenic bacteria in dead insects (e.g. Bacillus spp. and others)?
5. Do not repeat keywords present in title.
6. hundreds of acres? You mean millions of hectares surely?
7. There are better references on baculovirus formulation.
8. All death is "acute" I would say. You mean that they do not result in instantaneous or immediate death of the insect?
9. According to the producer's website, this would be equivalent to 3 x 10e6 OBs in 200 mL , or 1.5 x 10e4 OB/mL. Please mention this.
10. How were plants cleaned, just washed in water?
11. If subsequent experiments involved up to 250 virus-killed larvae, how were 4 replicates of 20 larvae sufficient to produce virus for subsequent bioassays? Please reword or explain.
12. Was the procedure performed for a group of 20 first instars, and another group of 20 second instars, etc.? Or were all instars present at the same time in mixed-instar groups? If the latter, how many of each instar were treated?
13. According to my calculations, this would be equivalent to 1.5 x 10e9 OBs in 20 L water, or 7.5 x 10e7 OBs/L. Please mention this.
14. Was a wetter-sticker or detergent used to improve the application of virus or insecticides? e.g. Tween 80, SDS, Triton X100?
15. At what stage of plant development did the virus and insecticide applications begin? How large were the plants at the moment of initial application?
16. If applications lasted 5 weeks, when did they happen during the experimental period of April to October?
17. Statistical analysis section should be moved to Methods.
18. F statistics required that treatment AND error degrees of freedom be reported, otherwise it is impossible to estimate the corresponding P-value.
19. Was virus-induced mortality observed in the controls? At what prevalence?
20. Remove the term "extract" from the entire manuscript
21. Table 1. Table 2. It is not appropriate to give percentage values to two decimal places when you had reduced sample sizes (less than 100 larvae). Suggest you present them to one decimal place.
22. Table 3. These are means but the damage scores during the period of treatment applications are the values that matter aren't they?
23a. You state that the yield was higher than the control in certain treatments but the statistical analysis indicates that they were similar, so you have no grounds to suggest that the plants in some treatments really produced more grain. This could have been a random effect (or so the statistical analysis indicates).
23b. Table 4. Did the value of the extra grain produced in the Escort treated plots cover the cost of 5 applications of the synthetic insecticide? Obviously Engeo and Littovir had no significant improvements on grain yields.
24. This text was already stated in the Introduction. Please delete.
25. This is not correct. Small larvae consume much much less than late instars. However, late instars are much more resistant to NPV infection as correctly stated on line 288. 
26. This is an important finding. The product Littovir (SpliNPV) did not kill more than 50% of the experimental insects even under optimum laboratory conditions. This may be because at least half of the laboratory insects were in the virus-resistant stages (L4 – L6), or the virus has low pathogenicity to S. frugiperda because it is the heterologous virus that is adapted to infect S. littoralis, not S. frugiperda. I think you should explain that this is probably not the best virus to use against S. frugiperda. It would be better to apply and recycle SfMNPV.
27. Another finding is that Engeo also provided rather poor control of S. frugiperda, isn't it?
28. Yes, clearly the timing of the application of treatments is critical to achieving suitable levels of plant protection. You need to highlight this.
29. Yes, delivery of active ingredients to the feeding site of the pest requires large volume applications (400 – 800 liters/ha), especially when the plants are in the whorl stage.
30. Yes, maize is fairly resistant to defoliation by S. frugiperda. The plants seem able to recover from damage during the whorl stage. This is not the case if the larvae attack the maize cobs of course. Farmers may not need to make repeated applications of costly synthetic insecticides except to protect maize cob development near the end of the crop cycle. The problem is that farmers get worried when they see obvious pest damage in the whorl stage.
31. You need to explain this more clearly. Insecticides don't increase the photosynthetic rate.
32. What do the large number of studies from Latin America indicate about NPV-based control using SfMNPV? If you research this, you may conclude that SpliNPV is a strange choice for S. frugiperda control in Kenya!
33. I disagree with this conclusion. I think your main finding is that Littovir provided only low mortality of S. frugiperda even when applied at very high rates (250 larvae). Perhaps a better conclusion would be "the idea of recycling was demonstrated with SpliNPV but might work considerably better with SfMNPV". Does that sound reasonable?
34. What about natural isolates of SfMNPV in Kenya? They could be found fairly easily I suspect by collecting naturally infected larvae from the field or even by collecting and processing soil samples from maize fields.
Minor details:
The English requires revision by the (presumably) native English-speaking author living in the UK.

Needs revision by the UK author.

Author Response

The authors would like to appreciate reviewer 2 for the comprehensive review, comments and suggestions. The input has greatly improved the manuscript.

Issue 1: Italicize Spodoptera frugiperda in the manuscript title and change the title.

Response: done line 2- Title changed to Management of Spodoptera frugiperda J.E. Smith using virus recycled virus inoculum from larvae treated with baculovirus under field conditions

Issue 2: In the manuscript summary remove the word extract from the sentences reading virus extract.

Response: Done lines 26, 28, 29, 30,32.

Issue 3: Change the word varied to variable.

Response: Done line 40.

Issue 4: The reviewer has suggested that the authors consider including a statement on the disadvantages of using of virus extracted from the FAW larvae for management of FAW.

Response: Due to limitation on the number of words in the abstract this statement has been included in the discussion. Lines 349-350.

Issue 5: Delete the words: virus extract, baculovirus, NPV.

Response: Deleted lines 48-49.

Issue 6: The authors did not understand what the reviewer meant or what should be done.

Response: No amendment done.

Issue 7: Replace the statement “after the affected cells” with the “infected cells”.

Response: Done lines 82-83.

Issue 8:  Rephrase the statement “cause acute death of the pest”.

Response: Rephrased, lines 87-88.

Issue 9: Change “6µl of Littovir in 200ml of water” to 5 x1011 OB/L.

Response: Done line 132.

Issue 10: The reviewer suggested rewriting of the sentence.

Response: Done, line 134.

Issue 11: The reviewer suggested rewriting of the sentence.

Response: Done, line 141.

Issue 12: Replace instar with instars.

Response: Done, line 137.

Issue 13:  Change 3ml in 20 to 5 x1011 OB/L.

Response: Done line 166.

Issue 14: The reviewer asked whether a detergent was used to clean the maize prior to feeding the FAW larvae.

Response: Only tap water was sued to clean the maize seedlings

Issue 15: At what stage of plant development did the virus and insecticide applications begin? How large were the plants at the moment of initial application.

Response: The treatment application began 4 weeks after germination and plants were about 30cm in height. The treatment application was done between May and June 2021.

Issue 16:  If applications lasted 5 weeks, when did they happen during the experimental period of April to October?

Response: The field experiment treatments were done between May and June 2021.

Issue 17: Move the section of statistical analysis to materials and methods and before results section.

Response: Done, line 227-231.

Issue 18: F statistics required that treatment AND error degrees of freedom be reported, otherwise it is impossible to estimate the corresponding P-value.

Response: The degrees of freedom (df) for the laboratory experiment is 4 since there were 5 treatments while for field experiment, the degrees of freedom (df) is 7 since there were 8 treatments.

Issue 19: Was virus-induced mortality observed in the controls? At what prevalence?

Response: When control mortality is used to correct for natural mortality, control mortality is not reported.

Issue 20: Remove the term "extract" from the entire manuscript.

Response: done

Issue 21: Table 1 and Table 2. It is not appropriate to give percentage values to two decimal places when you had reduced sample sizes (less than 100 larvae). Suggest you present them to one decimal place.

Response: This has been rectified in all the tables.

Issue 22: The reviewer asked whether the effect of time was evaluated in this study. Additionally, the reviewer asked that Table 3. These are means but the damage scores during the period of treatment applications are the values that matter aren't they?

Response: Suggested correction has been done and the authors agree with the reviewer that the damage score during the period of treatment application provide more information.

Issue 23: a. You state that the yield was higher than the control in certain treatments but the statistical analysis indicates that they were similar, so you have no grounds to suggest that the plants in some treatments really produced more grain. This could have been a random effect (or so the statistical analysis indicates).

Response: The Escort treatment was the only one that was significantly different from untreated control. The other treatments were similar to the untreated control. This has been rectified in the manuscript.

Issue 23b. Table 4. Did the value of the extra grain produced in the Escort treated plots cover the cost of 5 applications of the synthetic insecticide? Obviously Engeo and Littovir had no significant improvements on grain yields.

Response: The extra gain yield produced by Escort treated plots was able to cover the cost of application because the insecticide was purchased once and used for subsequent treatments. The authors agree that Engeo and Littovir had no significant grain yield improvement.

Issue 24: This text was already stated in the Introduction. Please delete..

Response: Done lines 288-290.

Issue 25: This is not correct. Small larvae consume much less than late instars. However, late instars are much more resistant to NPV infection as correctly stated on line 288.

Response: deleted lines 302-303.

Issue 26: This is an important finding. The product Littovir (SpliNPV) did not kill more than 50% of the experimental insects even under optimum laboratory conditions. This may be because at least half of the laboratory insects were in the virus-resistant stages (L4 – L6), or the virus has low pathogenicity to S. frugiperda because it is the heterologous virus that is adapted to infect S. littoralis, not S. frugiperda. I think you should explain that this is probably not the best virus to use against S. frugiperda. It would be better to apply and recycle SfMNPV.

Response: The authors agree with the reviewer, and this has been rectified in the discussion.

Issue 27: Another finding is that Engeo also provided rather poor control of S. frugiperda, isn't it?

Response: The authors agree with the observation by the reviewer that Engeo did not effectively control S. frugiperda .

Issue 28: Yes, clearly the timing of the application of treatments is critical to achieving suitable levels of plant protection. You need to highlight this.

Response: The authors agree with the reviewer that timing of application is critical in effective management of FAW. This has been highlighted in line 344-345.

Issue 29: Yes, delivery of active ingredients to the feeding site of the pest requires large volume applications (400 – 800 liters/ha), especially when the plants are in the whorl stage.

Response: The authors agree with the observation that made by the reviewer.

Issue 30:  Yes, maize is fairly resistant to defoliation by S. frugiperda. The plants seem able to recover from damage during the whorl stage. This is not the case if the larvae attack the maize cobs of course. Farmers may not need to make repeated applications of costly synthetic insecticides except to protect maize cob development near the end of the crop cycle. The problem is that farmers get worried when they see obvious pest damage in the whorl stage.

Response: The authors agree with the observations by the reviewer.

Issue 31: You need to explain this more clearly. Insecticides don't increase the photosynthetic rate.

Response: This has been edited line 360.

Issue 32: What do the large number of studies from Latin America indicate about NPV-based control using SfMNPV? If you research this, you may conclude that SpliNPV is a strange choice for S. frugiperda control in Kenya!

Response: Additional information has been added by comparing to previous studies in Brazil and Colombia.

Issue 33: I disagree with this conclusion. I think your main finding is that Littovir provided only low mortality of S. frugiperda even when applied at very high rates (250 larvae). Perhaps a better conclusion would be "the idea of recycling was demonstrated with SpliNPV but might work considerably better with SfMNPV". Does that sound reasonable?.

Response: The authors agree with the observation by the reviewer.

Issue 34: What about natural isolates of SfMNPV in Kenya? They could be found fairly easily I suspect by collecting naturally infected larvae from the field or even by collecting and processing soil samples from maize fields.

Response: The authors agree with the reviewer and the conclusion has been amended.

Round 2

Reviewer 2 Report

The authors have partially improved the manuscript but have failed to incorporate a number of important points. I have written numbered points on a scanned copy of the manuscript.

Numbered points:
1. L27. The authors emphasize that the yield in virus-treated plots was similar to that of one of the insecticides, but do not mention that it was also similar to the untreated control. Please be clear about this.
2. There is a very good article just published that the authors should mention which indicates that FAW defoliation was responsible for just 3% of variation in maize yield in Zambia. See Chisonga et al https://doi.org/10.1371/journal.pone.0279138
3. The authors state that they had removed the term "extract" from the manuscript, but its use persists at several points. Please replace this term. N.B: The authors also need to state clearly that they did not characterize the recycled inoculum, so that the identity of the virus that they applied to field plots is unknown and may have comprised SpliNPV or SfMNPV or a mixture of viruses.
4. L277. This sentence does not make sense.
5. The authors have not provided the error degrees of freedom that I previously requested for their F-statistics. F values are a ratio of within group and between group variation. Each F statistic therefore has a numerator and a denominator degrees of freedom, also known as the treatment and error degrees of freedom. If the authors struggle to understand this, they can consult numerous sources online. All F statistics should be presented with the treatment AND error degrees of freedom.
6. The authors state that they have adjusted percentage values to one decimal point. This has not been done in Table 1 or Table 2.
7. 1.5-fold??? According to my calculation this should be "a 1.8-fold higher yield in Escort-treated plots compared to the untreated control plots". Correct?
8. Here the authors emphasize that the yields in the baculovirus treatments did not differ significantly but should also mention that none of them increased yields above the untreated control.
9. This text seems to refer to sublethal effects of virus infection. However, the authors did not test for sublethal infection and the focus of the manuscript is on virus-induced mortality. They should remove this text.
10. This text reiterates the previous sentence. Please delete.
11. The authors have not mentioned the disadvantages of recycling inoculum that I previously requested. The most obvious disadvantages are: (1) The virus identity may change if inoculation results in activation of a covert infection (please cite a reference), (2) The insecticidal activity (quality) of the recycled inoculum may be reduced because the farmers are not familiar with identifying NPV-infected and other types of pathogens in larvae, (3) The larvae may harbor human pathogens and present a risk to the health of farmers (cite a reference).
12. Again, you need to make clear that the yields from virus plots were not significantly increased over those from the untreated control plots.
Finally, I would kindly request that the authors carefully read the manuscript for errors before they resubmit as I found the manuscript to be full of typos and obvious errors.

Needs editing.

Author Response

Authors would like to thank the reviewer for the useful comments. They have further improved the manuscript. All the comments have beed addressed in the responses to reviewer's comments. The report is here attached.

Issue 1: L27. The authors emphasize that the yield in virus-treated plots was similar to that of one of the insecticides, but do not mention that it was also similar to the untreated control. Please be clear about this.

Response: Done line 29.

Issue 2: There is a very good article just published that the authors should mention which indicates that FAW defoliation was responsible for just 3% of variation in maize yield in Zambia. See Chisonga et al https://doi.org/10.1371/journal.pone.0279138.

Response: The authors have read and cited the publication on effect of FAW defoliation on grain yield. Line 92-94

Issue 3: The authors state that they had removed the term "extract" from the manuscript, but its use persists at several points. Please replace this term. N.B: The authors also need to state clearly that they did not characterize the recycled inoculum, so that the identity of the virus that they applied to field plots is unknown and may have comprised SpliNPV or SfMNPV or a mixture of viruses.

Response: This has been done in the entire manuscript

Issue 4: L277. This sentence does not make sense.

Response: The sentence has been corrected.

Issue 5: The authors have not provided the error degrees of freedom that I previously requested for their F-statistics. F values are a ratio of within group and between group variation. Each F statistic therefore has a numerator and a denominator degree of freedom, also known as the treatment and error degrees of freedom. If the authors struggle to understand this, they can consult numerous sources online. All F statistics should be presented with the treatment AND error degrees of freedom.

Response: Error degrees of freedom have been provided as requested in all the ANOVA outputs.

Issue 6: The authors state that they have adjusted percentage values to one decimal point. This has not been done in Table 1 or Table 2.

Response: All the percentage values in tables 1 and 2 have been adjusted to one decimal point.

Issue 7: 1.5-fold??? According to my calculation this should be "a 1.8-fold higher yield in Escort-treated plots compared to the untreated control plots". Correct?

Response: This has been corrected. Line 290.

Issue 8: Here the authors emphasize that the yields in the baculovirus treatments did not differ significantly but should also mention that none of them increased yields above the untreated control.

Response: This has been rectified line 366.

Issue 9: This text seems to refer to sublethal effects of virus infection. However, the authors did not test for sublethal infection, and the focus of the manuscript is on virus-induced mortality. They should remove this text.

Response: The text has been deleted.

Issue 10: This text reiterates the previous sentence. Please delete.

Response: Deleted

Issue 11: The authors have not mentioned the disadvantages of recycling inoculum that I previously requested. The most obvious disadvantages are: (1) The virus identity may change if inoculation results in activation of a covert infection (please cite a reference) (2) The insecticidal activity (quality) of the recycled inoculum may be reduced because the farmers are not familiar with identifying NPV-infected and other types of pathogens in larvae, (3) The larvae may harbor human pathogens and present a risk to the health of farmers (cite a reference).

Response: This section has been revised and relevant references cited and added in the reference section

Issue 12: (a) Again, you need to make clear that the yields from virus plots were not significantly increased over those from the untreated control plots.
(b) Finally, I would kindly request that the authors carefully read the manuscript for errors before they resubmit as I found the manuscript to be full of typos and obvious errors.

Response: (a) This has been rectified in the manuscript.

(b) The manuscript has been revised by one of the co-authors -BL who is a native English speaker from UK.

Round 3

Reviewer 2 Report

I only have two issues with the modified manuscript.

1. Line 318 states "...some baculoviruses could pose human health risks". This is not true. Baculoviruses are safe for humans. As I stated previously, one of the risks of used recycled inoculum from virus-killed larvae is that larvae could harbor human pathogens such as human pathogenic bacteria that could potentially harm farmers.

2. The usual way to report degrees of freedom in F statistics is: F = 1.23; d.f. = 2, 34; P = 0.78.  However the authors use the term "edf" which I have never seen before. Please delete the term "edf" on lines 234, 243, 253, 255, 268 and report treatment and error degrees of freedom separated by a simple comma as I have indicated.

Thank you for addressing my previous concerns.

A few typos need correcting.

Author Response

The authors are grateful to the reviewer for the inputs and comments on the manuscript. Thee have further enhnaced the quality of the manuscript

Issue 1: 1. Line 318 states "...some baculoviruses could pose human health risks". This is not true. Baculoviruses are safe for humans. As I stated previously, one of the risks of used recycled inoculum from virus-killed larvae is that larvae could harbor human pathogens such as human pathogenic bacteria that could potentially harm farmers.

Response: This has been revised in the manuscript.

Issue 2: The usual way to report degrees of freedom in F statistics is: F = 1.23; d.f. = 2, 34; P = 0.78.  However, the authors use the term "edf" which I have never seen before. Please delete the term "edf" on lines 234, 243, 253, 255, 268 and report treatment and error degrees of freedom separated by a simple comma as I have indicated.

Response: The authors have correctly stated the degrees of freedom and error degrees of freedom as advised by the reviewer. Lines 244-245 249, 254, 264, 266, 279.